Forgetting faces over a week: investigating self-reported face recognition ability and personality

Kramer Robin S.S. remarknibor@gmail.com
School of Psychology, University of Lincoln , Lincoln , United Kingdom
Borghi Anna
Electronic publication date: 2021 Jul 16
Publication date: 2021
Volume: 9
Electronic Location ID: e11828
Received 2021 Feb 12; Accepted 2021 Jun 30
Copyright: ©2021 Kramer
Copyright year: 2021
Copyright holder: Kramer
License: This is an open access article distributed under the terms of the Creative Commons Attribution License, which permits unrestricted use, distribution, reproduction and adaptation in any medium and for any purpose provided that it is properly attributed. For attribution, the original author(s), title, publication source (PeerJ) and either DOI or URL of the article must be cited.
License URL: https://creativecommons.org/licenses/by/4.0/

Keywords: Face recognition, Long-term memory, Self-report, Individual differences, Personality

Funding: The authors received no funding for this work.

==============================
Background

Although face recognition is now well studied, few researchers have considered the nature of forgetting over longer time periods. Here, I investigated how newly learned faces were recognised over the course of one week. In addition, I considered whether self-reported face recognition ability, as well as Big Five personality dimensions, were able to predict actual performance in a recognition task.

Methods

In this experiment (N = 570), faces were learned through short video interviews, and these identities were later presented in a recognition test (using previously unseen images) after no delay, six hours, twelve hours, one day, or seven days.

Results

The majority of forgetting took place within the first 24 hours, with no significant decrease after that timepoint. Further, self-reported face recognition abilities were moderately predictive of performance, while extraversion showed a small, negative association with performance. In both cases, these associations remained consistent across delay conditions.

Discussion

The current work begins to address important questions regarding face recognition using longitudinal, real-world time intervals, focussing on participant insight into their own abilities, and the process of forgetting more generally.

Introduction

Although many researchers have argued that we are experts when it comes to perceiving and processing faces (e.g., Diamond & Carey, 1986), more recent evidence suggests that this expertise may be limited to familiar faces only (Young & Burton, 2018). Results have demonstrated that performance with familiar faces is significantly better in comparison with unfamiliar faces across a number of tasks, including recognition (Burton et al., 1999; Clutterbuck & Johnston, 2005; Ellis, Shepherd & Davies, 1979), sorting (Jenkins et al., 2011; Kramer et al., 2018), and matching (Bruce et al., 2001; Ritchie et al., 2015).

Despite the important role that familiarity plays in face perception, surprisingly little is known about the process of learning and familiarisation. Early studies emphasised the duration or frequency of encounters (e.g., Shapiro & Penrod, 1986), although this work had limited success in providing a better understanding of the underlying mechanisms, perhaps due to their reliance on highly standardised images. Recent research has shown that images of the same person can appear very different (Jenkins et al., 2011), and that this within-person variability is itself idiosyncratic (Burton et al., 2016). As such, increasing exposure to the different appearances of a single face aids learning and subsequent recognition of that face (Andrews et al., 2015; Ritchie & Burton, 2017), presumably through the generation of a robust internal representation (Burton et al., 2005). In contrast, limiting exposure to variability results in a greater reliance on image-level properties (Hancock, Bruce & Burton, 2000), causing difficulties when later generalising to new instances.

While researchers are beginning to understand how learning and familiarisation can occur over time and with exposure to a new face, few studies have considered the inverse process: how faces are forgotten. Although several properties of the initial learning experience play an important role (e.g., duration of exposure; for a review, see Deffenbacher et al., 2008), evidence has also identified individual differences that influence forgetting, such as the level of stress felt when a face is learned in an eyewitness context (Deffenbacher et al., 2004). It is likely that there are also more stable differences across individuals that relate to the nature of face forgetting, given the strong genetic (heritable) basis underlying face recognition ability (Shakeshaft & Plomin, 2015; Wilmer et al., 2010).

From the perspective of police or security recruiters who utilise, for example, an employee’s ability to recognise a face learned previously, it is important to determine whether there are any easily measured predictors regarding performance. For example, individual differences related to personality domains may be one such candidate (e.g., Lander & Poyarekar, 2015), perhaps resulting from the above-mentioned genetic underpinnings of ability. Another could be an individual’s self-insight (e.g., Matsuyoshi & Watanabe, 2021), making the selection of workers far simpler if each was aware of his or her own abilities. However, as yet, no research has considered how these factors may interact with the process of forgetting. Those who self-report as demonstrating higher abilities with face recognition may be correct when faces were learned only minutes ago, but such insights may be misplaced when targets have to be remembered over the longer term. The same interaction could also be present for personality domains, where specific traits are more strongly associated with face recognition (Lander & Poyarekar, 2015; Li et al., 2010), or memory performance (Stephan et al., 2020), but the relationship between these three factors remains untested. The current experiment, therefore, will investigate the process of face forgetting over longer time periods than are usually considered in research designs, and will also begin to explore whether individual differences predict face recognition ability over different retention intervals.

Forgetting over time

While numerous studies have focussed on face recognition, these have typically featured little or no delay between learning and test (e.g., Baker, Laurence & Mondloch, 2017; Duchaine & Nakayama, 2006; Lander & Davies, 2007; Ritchie & Burton, 2017; Rule, Slepian & Ambady, 2012; Russell, Duchaine & Nakayama, 2009; Zhou et al., 2018). Of course, real-world recognition almost always involves some form of delay, which can often extend over many years. For this reason, Bahrick and colleagues (1975) used a cross-sectional design in order to investigate retention intervals of up to 57 years by exploring participants’ recognition of yearbook photographs. Their findings suggested that under conditions of prolonged acquisition (i.e., during participants’ high school education), information was preserved for much longer than laboratory demonstrations might show. Of course, the insights gained through this type of design were at the expense of experimental control over several variables.

A more recent study sought to balance the naturalistic learning and forgetting of faces over several years with control over important factors that affect familiarity (Devue, Wride & Grimshaw, 2019). The researchers recruited participants who had watched all six seasons of the TV series Game of Thrones, and subsequently tested their recognition across a variety of main and supporting characters. Interestingly, although there were clear benefits due to increased and more recent exposure, even well-learned faces were forgotten over time. In addition, the alteration of external features (e.g., hair colour or accessories) led to a decrease in recognition for even the most familiar faces, reiterating the findings mentioned earlier regarding the substantial effect of within-person variability in appearance.

Despite the logistical difficulties involved with incorporating delays into experimental designs, a number of studies have provided evidence of recognition after longer term intervals. For example, Davis & Tamonyte (2017) asked participants to learn a target from a 1-min video clip and subsequently identify the individual from a nine-person video line-up which took place approximately ten days later. Accuracy on this task (in the ‘no disguise’ condition) was moderate and depended on whether the target was present (33%) or absent (80%) in the line-up. Other researchers have also employed longer delays across a variety of face recognition tasks (e.g., 28 days –Courtois & Mueller, 1981; 23 days –Sauer et al., 2010; 35 days –Shepherd & Ellis, 1973; 1 month –Shepherd, Gibling & Ellis, 1991; 4 months –Shepherd, Ellis & Davies, 1982; 30 days –Yarmey, 1979), with a meta-analysis of these studies finding a small to medium association between longer retention intervals and face forgetting (Deffenbacher et al., 2008).

Most relevant to the current work, Davis and colleagues (2020) constructed a face learning task in order to investigate recognition after variable delay intervals. Each participant viewed ten 1-min video clips depicting the face and upper body of unfamiliar individuals, and was subsequently tested on their recognition of these ten actors using six-person target-present video line-ups. Importantly, the delay between learning and test varied from 1–182 days, representing a significant period over which forgetting would occur. For the shortest delay (1–6 days), hits rates were already low (0.46), decreasing even further (0.26) for the longest delay group (56+ days). In a second experiment, twenty unfamiliar individuals were learned using 30 s video clips, with both target-present and target-absent photo line-ups presented at test. Retention intervals varied from almost immediately to 50 days after learning. Even for those participants who were tested within one day of learning, performance was poor (hits = 0.52, correct rejections = .32), and for those who were tested after 28 days or longer, performance had decreased even further (hits = 0.27, correct rejections = .34). It is clear from these results that the learning tasks were highly difficult and showed increased forgetting over time, with retention intervals showing medium-sized associations with hit rates in both experiments (Experiment 1: −0.31; Experiment 2: −0.43).

Although charting the progression of forgetting over longer periods of time is informative, the majority of forgetting takes place during the first 24 hours (Deffenbacher et al., 2008). The forgetting curve (Ebbinghaus, 1885) that has come to describe this deterioration over time is best modelled by a power or exponential curve (Averell & Heathcote, 2011; Rubin & Wenzel, 1996; Wixted & Ebbesen, 1991), with the steep early decline suggesting that this initial period should be the focus of further exploration.

Self-report measures and face recognition ability

In recent years, researchers have increasingly focussed on whether people have accurate insights into their face recognition abilities. Put simply, are participants aware of how well they perform on such tasks? Although originally devised as a method of screening adults for developmental prosopagnosia, scores on the 20-item prosopagnosia index (a measure of self-reported ability; Shah et al., 2015a) have since demonstrated medium-sized associations with performance on the Glasgow Face Matching Test (Burton, White & McNeill, 2010) and the Cambridge Face Memory Test (CFMT; Duchaine & Nakayama, 2006) in the general population (Gray, Bird & Cook, 2017; Livingston & Shah, 2018; Shah et al., 2015b; Ventura, Livingston & Shah, 2018). These results suggest that participants do indeed have some level of meta-cognitive insight into their own abilities. Similarly, scores on a 15-item questionnaire developed in a Hong Kong population for the screening of congenital prosopagnosia (HK15; Kennerknecht, Ho & Wong, 2008) have since shown small associations with the CFMT in typical adults (Palermo et al., 2017). Interestingly, after removal of four dummy questions that were irrelevant with respect to face recognition, the resulting 11-item subset of questions (HK11) showed a medium-sized association with an East Asian version of the CFMT (Matsuyoshi & Watanabe, 2021). Although these questionnaires are those most frequently utilised as a measure of insight, other tools have also been developed with some success (Arizpe et al., 2019; Bobak, Mileva & Hancock, 2019; Saraiva et al., 2019). Taken together, it seems clear that, to some extent, people are aware of their face recognition abilities.

Problematically, such measures of insight into ability have only been correlated with tests of face recognition without delays (e.g., the CFMT). There is no evidence that self-reported abilities show an association with performance when the task involves learning and then later recognition following a delay. In a recent pre-print (Davis et al., 2019), researchers included a single-item measure of insight in order to determine whether participants believed they were below or above average in face recognition ability (using a 1–5 scale) and investigated the recognition of learned identities up to six months later (see also Davis et al., 2020). However, this measure of insight failed to predict accuracy on the task. Of course, this may be due to the use of a single item for quantifying insight, and the question remains as to whether better established measures are able to predict performance in this domain.

Related, there is some evidence that self-report measures of personality may also be associated with performance on face-related tasks, although results appear to be mixed. For example, extraversion may be related to face recognition (Lander & Poyarekar, 2015; Li et al., 2010), while face matching does not appear to be related to personality measures, other than perhaps facets of neuroticism (anxiety—Megreya & Bindemann, 2013; no associations—Lander & Poyarekar, 2015). More recently, no relationship was found between personality factors and measures of face memory and matching (McCaffery et al., 2018). When searching for faces in crowds, there is also evidence that personality may (Kramer, Hardy & Ritchie, 2020) or may not (Davis et al., 2018) be associated with performance measures. Again, there is a lack of research investigating whether personality facets are related to performance on a recognition task over a time delay.

The current experiment

In light of the unanswered questions highlighted above, I identified two aims for the current experiment, which can be summarised as follows. First, little is known regarding the nature of face forgetting over the first 24 hours post-learning. As such, this study focussed on the first week following exposure to these new identities, but with the specific goal of understanding this crucial, initial time period. In order to learn more about real-world forgetting, participants learned faces from short video interviews, comparable with exposure during a brief conversation. Second, although several studies have now found evidence that self-reported face recognition abilities, and personality measures to a lesser extent, were moderately predictive of actual performance, the tasks employed in those experiments did not include any form of delay between learning and test. As such, it remains unclear as to whether insight into one’s own ability and personality are predictive when incorporated into a more ecologically valid design, and whether any such associations are altered by the retention interval.

Method

Participants

After restricting eligibility to those located in the USA, the UK, Australia, Canada, and New Zealand (unless otherwise specified—see below), where the majority of residents speak English, 2,085 participants were recruited through Amazon Mechanical Turk (MTurk). Of these, 570 (231 women; age M = 39.2 years, SD = 11.8 years; 78.4% self-reported ethnicity as White) completed the full study (both learning and test sessions), correctly answered all attention checks, and were unfamiliar with all of the identities used as stimuli. Table 1 provides a summary for each condition in terms of sample sizes, attrition rates, and exclusions. Participants who completed the learning session were paid US $0.75, and those who completed the test session received a further US $0.75. Combined, this wage equated to approximately $6 per hour, although the second test session took less time but was priced equally in order to encourage participants to complete both sessions. No payment was given for a session in which attention checks were answered incorrectly. Participants provided informed consent online prior to taking part, and received an online debriefing upon completion, in line with the university’s ethics protocol. Ethical approval for this experiment was granted by the University of Lincoln’s ethics committee (ID 3508).

Table 1 Summary of sample size and exclusion information for each condition.

		Delay	
		None	6 h	12 h	1 day	7 days	
Learning	Completed	235	478	571	400	401	
	Excluded - attention checks	63	140	140	144	130	
	Excluded - familiarity check	–	59	69	53	46	
	Final sample	–	279	362	203	225	
Testing	Completed	–	119	110	161	165	
	Excluded - attention checks	27	6	5	19	14	
	Excluded - familiarity check	32	11	5	16	22	
	Final sample	113	102	100	126	129	

MTurk is a platform well-suited to longitudinal research (Chandler & Shapiro, 2016; Cunningham, Godinho & Kushnir, 2017), and retention rates here (see Table 1), as with other studies (Shapiro, Chandler & Mueller, 2013; Stoycheff, 2016), were relatively high. In order to maximise the likelihood that those who participated in the learning session would return to complete the test session, MTurk workers who had completed the first session were contacted individually using the ‘pyMTurkR’ package (Burleigh & Leeper, 2020) and invited to complete the second session.

An a priori power analysis was conducted using G*Power 3.1 (Faul et al., 2007), based on the correlation between self-reported face recognition abilities (using the same questionnaire featured here) and actual performance in previous work (−.38—Matsuyoshi & Watanabe, 2021). In order to achieve 80% power at an alpha of .05, a total sample size of 49 was required. As such, I aimed to recruit a minimum of 49 participants (after exclusions) in each delay condition (described below).

Materials

From a larger database of 80 White (non-UK) European individuals (predominantly German or Dutch), ten identities (four women) were chosen to serve as faces to be learned in the current experiment. This selection was based upon the performance of a different sample of participants, where these identities were chosen to represent a range of face memorability scores, although this was not investigated here. All were identified as nationally well-known (e.g., singers, actors, athletes) while none had reached international levels of fame. For each person, a video interview was found on YouTube in which they were filmed for at least one minute using a fixed camera (i.e., the viewing angle of their face remained unchanged throughout) and spoke in a language other than English (simulating natural conversation without the audio content providing additional information that might aid learning). In all cases, the interviewer was positioned close to the camera, resulting in a view of the face that was relatively front-on.

For each identity, a continuous 30 s segment was selected from the initial YouTube video in which the person was predominantly front-on and speaking for most or all of the time (rather than simply listening to the interviewer). The video was also cropped to 350 x 350 pixels in order to include only the head and the top of the shoulders (and the background contained within that frame; see Fig. 1). These videos were in colour and included the audio information.

Figure 1 Images of the same identity, representative of the videos presented during the learning task (left) and images presented during the recognition test (right).

Photo credits: Robin Kramer.

In order to create the recognition test, 20 additional identities (11 women) from the original database were chosen at random with the caveat that half of the final set of 30 identities were women. For each of these 30 people, a high-quality, colour photograph was downloaded from Google Images in which they were approximately forward-facing. For the ten identities to be learned, these photographs were chosen so that their appearance resembled that of the videos in which they featured, e.g., matched for age, hair style, facial hair and glasses where applicable, etc. Importantly, these were images taken in new contexts in all cases, and were not still frames from the videos. Images were subsequently cropped to 350 × 350 pixels, displaying only the head and the top of the shoulders (and the background contained within that frame; see Fig. 1).

In order to measure participants’ self-reported face recognition abilities, I used the HK15 questionnaire (e.g., “it takes me a long time to recognise people”; Kennerknecht, Ho & Wong, 2008). For each item, participants select a response from the following: strongly agree, agree, uncertain, disagree, strongly disagree. After reverse coding eight items, overall score is calculated by summing individual responses, with lower scores indicating higher self-reported estimates of face recognition ability. Subsequent removal of four dummy questions that are irrelevant with respect to face identity recognition (e.g., “I get lost in new places”) produces an 11-item subset of questions (HK11; score range 11-55) which has previously shown a medium-sized association with actual face recognition performance (r =  − 0.38; Matsuyoshi & Watanabe, 2021). The HK11 demonstrates high levels of reliability (Cronbach’s α = 0.84; Matsuyoshi & Watanabe, 2021).

In order to measure participants’ self-reported Big Five personality domains, I used the Ten-Item Personality Inventory (TIPI; Gosling, Rentfrow & Swann Jr, 2003). For each item, participants respond using a 1 (disagree strongly) to 7 (agree strongly) Likert scale. Participants are instructed to rate the extent to which pairs of traits apply to them, e.g., “I see myself as extraverted, enthusiastic.” After reverse coding five items, the overall score on each domain is calculated by averaging the two individual responses, with higher scores indicating higher self-perceived applicability for that domain. This questionnaire is a short measure when compared with most personality inventories, but strong correlations have been shown between the TIPI dimensions and the well-validated 60-item NEO-PI-R (Costa Jr & McCrae, 2008; Ehrhart et al., 2009), as well as the 40-item EPQ-R (Eysenck & Eysenck, 1993; Holmes, 2010). With only two items per scale, the TIPI demonstrates low reliability (Gosling, Rentfrow & Swann Jr, 2003), with Cronbach’s α values of 0.68 (extraversion), 0.40 (agreeableness), 0.50 (conscientiousness), 0.73 (emotional stability), and 0.45 (openness to experience). However, scales with small numbers of items commonly show low alpha scores (Gosling, Rentfrow & Swann Jr, 2003) and so test-retest reliability (r = .72 over a six-week span) is considered a more appropriate measure of an instrument’s quality. Therefore, while providing a similar measure to longer inventories, the benefit of its use here is its minimal demands on participant time, requiring approximately one minute to complete.

Procedure

The experiment was completed using the Gorilla online testing platform (Anwyl-Irvine et al., 2020). I collected information regarding the participant’s age, gender, and ethnicity, as well as their MTurk Worker ID. By assigning a ‘qualification’ using this Worker ID, I was able to associate data files across learning and test sessions, as well as prevent participants from taking part in more than one delay condition. These conditions comprised no delay, six hours, twelve hours, one day, and seven days, with assignment to condition described below.

Participants first completed the TIPI and HK15 questionnaires in order that their experience with the recognition task did not affect their self-estimates of ability. Next, they were shown the ten 30 s videos in a random order and instructed, “Please watch the videos carefully and learn to recognise each person’s face.” Participants were also asked to view the videos with the sound enabled in order to make the learning experience more natural, although they were not expected to understand what was being said (given that the spoken language was not English). A ‘play video’ button took participants to a new screen where the video started playback for each learning trial, allowing participants to control their progress. However, once started, videos could not be paused, rewound, or replayed.

Two attention checks were included during learning, appearing before the fourth and eighth video presentations, given that attentiveness is a common concern when collecting data online (Hauser & Schwarz, 2016). Each of these two trials instructed the participant to click on either the ‘left’ or ‘right’ button presented onscreen. For instance, “Attention Check: Please click the LEFT button now (in less than 10 s) to show you’re paying attention” was displayed onscreen. By requiring participants to respond within this limited time window, I could identify those who were not paying attention or may have started videos and then pursued other activities.

For participants in the ‘no delay’ condition, the learning task was immediately followed by the recognition test. Participants were presented with the 30 test images and asked to decide whether the face was seen during learning or not. Responses were provided using a labelled rating scale: (1) I’m sure it’s someone I learned; (2) I think it’s someone I learned; (3) I don’t know; (4) I think it’s someone I didn’t learn; (5) I’m sure it’s someone I didn’t learn.

Two additional trials were also included as attention checks during the recognition test. Each of these two trials consisted of a celebrity’s photograph (not one of the original 30), similar in appearance to a real trial (background present, identical image size). However, the internal features of the face were replaced with text, instructing the participant to respond with either ‘2’ or ‘4’ on the response scale. For instance, “Attention Check: Please respond with ‘2’ here.” By requiring participants to give different responses across the two attention checks, I could identify those who were not paying attention or clicked the same button onscreen throughout the experiment irrespective of what was being displayed.

The presentation order of the 32 images was randomised for each participant. Responses were given using the mouse and were self-paced. Finally, participants were asked how many (if any) of the faces they had recognised from their experiences prior to the experiment.

For participants in conditions with a delay between the learning and test sessions, the familiarity question directly followed learning (with no test included). During the separate test session, participants completed demographic information again (but did not repeat the two questionnaire measures), followed by the recognition test and then another familiarity question.

Regarding knowledge about a subsequent test, all participants were informed onscreen at the start of the learning task that there would be a recognition test afterwards. However, for those in conditions with a delay, participants were simply told that this test would take place “in the next several weeks” and were therefore unaware of the specific length of delay to which they were assigned. For logistical reasons (e.g., making the experiment available for certain periods of the day, keeping track of completions and inviting the appropriate participants to the test session, etc.), rather than randomly allocating participants to conditions, recruitment for the five delay conditions took place in the following sequence: no delay, one day, seven days, six hours, twelve hours.

For the ‘one day’ and ‘seven days’ conditions, both the learning and test sessions were each made available for approximately 24 hours or until no additional MTurk workers had taken part for approximately two hours. In all conditions, if a sufficient sample size had not been reached then the process of recruitment for both sessions was repeated as necessary. As mentioned above, for the test sessions, qualifying participants were notified via email as the session was posted on MTurk. However, for the ‘six hours’ and ‘twelve hours’ delays, both the learning and test sessions were made available for only 1.5 hours each. Again, qualifying participants were notified as the test session was made available and, for these conditions, were informed that it would only be accessible for the next hour and a half. In order to avoid participants completing the learning session who would not be available to complete the test session six or twelve hours later due to the time of day (i.e., requiring one session to take place during the night), MTurk workers from Australia and New Zealand were excluded from recruitment for these two conditions only (given the difference in time zones between these two countries and the other three). In addition, for the ‘twelve hours’ condition, session timings were chosen in order to avoid including a night’s sleep.

Participants were prevented from completing the experiment using mobile phones (via settings available in Gorilla) in order to ensure that videos and images were viewed at an acceptable size onscreen.

Results

Data analysis included only those participants who correctly answered all attention checks and reported being unfamiliar with all of the identities used as stimuli (see Table 1).

For each participant, I calculated the hit and false alarm rates for each possible threshold (i.e., the theoretical boundary between ‘learned’ and ‘new’) along the recognition response scale (1 through 5). Rather than making explicit judgements about whether identities were learned or new, participants rated the likelihood that each identity had been learned. This approach, therefore, focussed on their internal representation of this likelihood (a continuous measure) rather than forcing a binary decision based on an internal threshold that differentiates a ‘learned’ from a ‘new’ identity. Plotting these values produced the receiver operating characteristic (ROC), with the area under this ROC curve (AUC) representing a measure that is widely used to assess the performance of classification rules over the entire range of possible thresholds (Krzanowski & Hand, 2009). As such, AUC allowed quantification of the performance of a classifier (here, each participant), irrespective of where the cut-off between binary ‘learned’/’new’ responses might have been placed. This more fine-grained analysis bypassed the need to rely on a participant’s final decision (‘learned’/’new’) in favour of investigating what was presumably the underlying perception—the likelihood that this identity was someone previously learned. These data are summarised in Table 2, along with descriptive statistics for the questionnaire responses. As Table 2 illustrates, the current sample scored lower on both extraversion and openness in comparison with population norms (extraversion = 4.44, openness = 5.38; Gosling, Rentfrow & Swann Jr, 2003), while HK11 scores were similar to those reported in the original study (M = 24.04; Matsuyoshi & Watanabe, 2021).

In order to investigate whether performance (AUC) differed across the five delay conditions, I carried out a univariate analysis of variance (ANOVA), which showed a statistically significant main effect of delay, F (4, 565) = 23.60, p < .001, η2p = 0.14. Pairwise comparisons (Bonferroni corrected) revealed that the shortest conditions (no delay, 6 h, 12 h) all differed from the longest conditions (1 day, 7 days; all p s < .001). However, I found no differences within these two subcategories (shortest delays: all p s > .086; longest delays: p = 1.00). The five conditions are shown in Fig. 2.

Table 2 Summary data for participants’ responses.

Condition	Delay (hours)	AUC	HK11	E	A	C	ES	O	
No delay	0	0.75 (0.16)	24.42 (7.94)	3.77 (1.49)	4.89 (1.27)	5.15 (1.38)	4.62 (1.46)	5.02 (1.24)	
6 h	6.03 (0.39)	0.70 (0.16)	22.57 (7.39)	3.50 (1.67)	5.10 (1.31)	5.68 (1.20)	4.85 (1.49)	4.85 (1.40)	
12 h	12.28 (0.54)	0.73 (0.12)	22.31 (6.48)	3.47 (1.56)	5.03 (1.40)	5.73 (1.13)	4.97 (1.42)	4.87 (1.25)	
1 day	32.17 (7.05)	0.62 (0.14)	24.44 (7.01)	3.88 (1.41)	4.99 (1.18)	5.48 (1.29)	4.89 (1.26)	4.98 (1.24)	
7 days	173.26 (5.68)	0.60 (0.14)	23.70 (7.40)	3.72 (1.62)	5.09 (1.25)	5.47 (1.33)	4.93 (1.50)	4.89 (1.26)	
All participants	49.55 (68.02)	0.67 (0.15)	23.56 (7.30)	3.68 (1.55)	5.02 (1.27)	5.49 (1.29)	4.85 (1.42)	4.92 (1.28)	
Notes.

E Extraversion

A Agreeableness

C Conscientiousness

ES Emotional Stability

O Openness

Values are presented as M (SD).

Figure 2 The effect of delay on face recognition performance.

The dashed line depicts a power model for this relationship. Error bars represent 95% confidence intervals.

Next, I considered whether self-reported face recognition ability (HK11) and Big Five personality domains were associated with performance (AUC) across the whole sample. Correlations with AUC were as follows: HK11, r (568) = -.34, p < .001; extraversion, r (568) = -.12, p = .004; agreeableness, r (568) = .17, p < .001; conscientiousness, r (568) = .17, p < .001; emotional stability, r (568) = .09, p = .027; and openness, r (568) = .12, p = .006. However, after applying Bonferroni correction for multiple tests, the correlation with emotional stability was no longer statistically significant.

I then investigated whether self-reported face recognition ability and personality domains predicted performance after controlling for differences as a result of delay condition. To this end, I carried out a hierarchical linear regression, including delay condition (reference category: no delay) as the initial predictor (replicating the above ANOVA). For each of the six additional predictors (HK11 scores and the five personality domain scores), I compared the model in which it was included to the previous model in which it was absent (using the anova function in R). If the model’s improvement was statistically significant, this process was repeated in order to consider the inclusion of the predictor’s interaction with delay condition. This process, along with the final model, F (6, 563) = 35.29, p < .001, R2 = 0.27, can be seen in Table 3, where only HK11 and extraversion were included (Step 3). All other predictors (agreeableness, conscientiousness, emotional stability, openness) and their interactions with delay failed to significantly improve the model (all p s > .05). As such, the relationships between HK11 and AUC, as well as extraversion and AUC, were shown to be consistent across delay conditions. As Table 3 illustrates, HK11 scores were a stronger predictor of performance in comparison with extraversion. In addition, although the inclusion of extraversion significantly improved the fit of the model, the increase in R2 (0.02) was small.

Table 3 The hierarchical regression analysis for predicting performance (AUC).

	Variable	B	SE	β	t	R2	ΔR2	
Step 1						0.14	0.14	
	Intercept	0.75	0.01		55.33***			
	Delay: 6 h	−0.05	0.02	−0.13	−2.63**			
	Delay: 12 h	−0.02	0.02	−0.05	−1.12			
	Delay: 1 day	−0.13	0.02	−0.34	−6.85***			
	Delay: 7 days	−0.14	0.02	−0.39	−7.81***			
Step 2						0.25	0.11	
	Intercept	0.92	0.02		40.41***			
	Delay: 6 h	−0.06	0.02	−0.16	−3.52***			
	Delay: 12 h	−0.04	0.02	−0.09	−2.00*			
	Delay: 1 day	−0.13	0.02	−0.34	−7.33***			
	Delay: 7 days	−0.15	0.02	−0.41	−8.66***			
	HK11	−0.01	0.00	−0.33	−9.12***			
Step 3						0.27	0.02	
	Intercept	0.98	0.03		35.78***			
	Delay: 6 h	−0.07	0.02	−0.17	−3.80***			
	Delay: 12 h	−0.04	0.02	−0.10	−2.29*			
	Delay: 1 day	−0.13	0.02	−0.34	−7.34***			
	Delay: 7 days	−0.15	0.02	−0.41	−8.82***			
	HK11	−0.01	0.00	−0.35	−9.61***			
	Extraversion	−0.01	0.00	−0.14	−3.90***			
Notes.

Delay reference category = no delay.

* p < .05.

** p < .01.

*** p < .001.

Finally, Fig. 2 illustrates the forgetting curve (Ebbinghaus, 1885) for these data, generated using MATLAB’s fit function (model fit, R2 = 0.81). The model includes a vertical shift, as well as a horizontal shift in order to allow for a delay of zero (Averell & Heathcote, 2011; Wixted & Ebbesen, 1991). Power functions are generally accepted as suitable models for forgetting, in particular when averaging across participants and therefore focussing on group-level performance (Averell & Heathcote, 2011; Murre & Chessa, 2011; Wixted & Ebbesen, 1991).

Discussion

The experiment presented here was designed with two main aims. First, I investigated whether self-report measures were predictive of face recognition abilities, even when learning and test were separated by a delay. Second, I was interested in mapping the general process of forgetting over time, with particular focus on the first 24 hours after initial exposure.

Recent research has found a medium-sized association between self-reported ability (using the HK11 questionnaire) and actual face recognition performance (Matsuyoshi & Watanabe, 2021). Indeed, several researchers have demonstrated similar-sized correlations between various self-report instruments and different measures of performance (Arizpe et al., 2019; Bobak, Mileva & Hancock, 2019; Gray, Bird & Cook, 2017; Livingston & Shah, 2018). Here, I demonstrated that this association remained when delay intervals were introduced. Indeed, the relationship between participants’ self-reported abilities and their actual performance was constant across these different delays. This is an important result since the only previous study to consider this issue (Davis et al., 2019) found no evidence of insight after delays, although the authors acknowledged that this may have been due to the use of a single item to quantify insight.

In the current work, I found that HK11 scores provided a measure of self-reported ability that was moderately predictive of performance, no matter whether recognition was required immediately or after a delay of up to seven days. That the relationship remained constant across the different delays is both novel and interesting, given that previous research has not considered the possibility that the accuracy of self-report measures may be dependent on the interval between learning and test. Clearly, memory requirements varied across the delay intervals used, and these findings suggest that participants may incorporate this notion of recognising faces over unspecified amounts of time into their responses. Indeed, HK11 items do not refer to specific time delays and so it might be interesting to consider whether self-report measures that do specify the interval under consideration (i.e., asking only about recognition abilities a week after meeting someone) could lead to more accurate insights regarding the specific delay in question, although this remains to be investigated.

Regarding real-world applications, it is important that measures of insight extend beyond tests of immediate recognition since screening individuals for their abilities, for example, would almost certainly be with the intention of employing their skills in contexts involving substantial delays (Davis et al., 2016). It is likely that the requirements involved in learning a face, and then having to recognise that face immediately afterwards, are very different from those whereby targets are recognised weeks or months later, as were the conditions faced by police officers investigating the 2011 London riots (Davis, 2019), for example.

In addition to self-reported abilities, there is some evidence to suggest that particular facets of personality may be associated with those who perform better on face-related tasks. Extraversion may be one such candidate, showing associations with abilities in both face recognition (Lander & Poyarekar, 2015; Li et al., 2010) and spotting faces in crowds (Kramer, Hardy & Ritchie, 2020), although such evidence is far from conclusive (Davis et al., 2018; McCaffery et al., 2018). Here, I found a small association between this dimension and recognition performance. However, while the association remained constant across the five delay conditions, it was somewhat surprising that extraversion showed a negative correlation with performance (although see Kramer, Hardy & Ritchie, 2020). Clearly, there is a need to investigate this result further since it seems counterintuitive that introverted people may perform better with learning and later recognising faces. One explanation is that extraversion itself may comprise two subcomponents (Bornstein, Hahn & Haynes, 2011; Roberts, Walton & Viechtbauer, 2006): social dominance (surgency, assertiveness) and social vitality (sociability, fun-seeking). For this reason, an overall measure of this dimension could be difficult to interpret and might mask different underlying associations with recognition ability. An additional issue to note is that, as a subsample, MTurk workers are typically more introverted than the general population (Burnham, Le & Piedmont, 2018), which could limit the conclusions drawn from experiments recruiting from this particular participant pool. Indeed, Table 2 suggests that the current sample scored lower on both extraversion and openness in comparison with population norms (Gosling, Rentfrow & Swann Jr, 2003). Therefore, although personality does appear to predict face recognition ability in this experiment, I recommend further research in order to address this issue more conclusively.

This experiment also aimed to explore the forgetting of faces longitudinally after realistic learning. To this end, I utilised short video interviews where individuals were speaking and facing towards the camera in order to simulate a brief real-world encounter. Previous research has shown that the process of forgetting typically follows a power or exponential curve (e.g., Averell & Heathcote, 2011; Rubin & Wenzel, 1996; Wixted & Ebbesen, 1991), with a steep early decline. However, no research to date has explicitly investigated the forgetting curve associated with faces and its formulation. While Deffenbacher and colleagues (2008) attempted to model forgetting functions using several datasets, the authors were forced to estimate performance after ‘no delay’ due to a lack of data and fitted their functions “by eye” (p. 145). Importantly, these earlier experiments involved learning faces under conditions that failed to mirror the real world (e.g., through the use of the same static images at learning and test). In line with general predictions regarding forgetting, my results revealed that performance fell dramatically within the first 24 hours. In addition, the deterioration between those tested in the first 12 hours and those tested after 24 hours was also significant. After this point, no further deterioration was seen during the subsequent 1–7 day period.

An interesting issue to consider, although beyond the remit of the current work, is the function of sleep for those who participated in the ‘1 day’ and ‘7 days’ conditions. (The timings of the ‘12 hours’ condition were chosen to avoid including a night’s sleep for participants, who were restricted to the USA, the UK, and Canada.) While previous work has suggested that face learning may benefit from memory consolidation during sleep (Wagner et al., 2007), more recent research has argued that, instead, it is wakefulness during retention that diminishes memory for faces (Sheth, Nguyen & Janvelyan, 2009). Ongoing sensory stimulation interferes with visual memory while sleep shelters the individual from this interference. Although the current findings are in line with previous research on sleep and wakefulness effects, further work might incorporate this factor in order to investigate face forgetting during this first 24-hour period, e.g., by equating retention intervals while manipulating the presence/absence of sleep.

In order to simulate a brief real-world encounter while minimising the influence of the content of the conversation on learning/remembering, videos were constructed in which identities spoke in languages other than English (predominantly German or Dutch), while participants were recruited from countries where English is the primary language (the USA, the UK, Australia, Canada, and New Zealand). However, it is possible that a small number of participants understood what one or more of the targets were saying, and conversely, it may be that some participants did not have the sound enabled during the task (although they were instructed to do so). While previous research has demonstrated the beneficial role of motion in learning new faces (Lander & Bruce, 2003; Lander & Davies, 2007), to my knowledge, there is no research investigating whether the presence of speech aids face learning. In the current experiment, enabling sound during learning may simply have better captured participants’ attention, although it is possible that an additional understanding of what the identities were saying (while likely infrequent due to recruitment restrictions) could have helped with learning those particular faces. Even so, future research may consider whether the inclusion of additional information learned through speech could benefit learning and later recognition.

Regarding the time taken to forget, it is worth noting that the identities used here were national celebrities, chosen for logistical reasons—the availability of both naturalistic images and video interviews. As such, it may be the case that these people were not representative of the general population in terms of attractiveness and/or distinctiveness, both of which are known to affect face memory (e.g., Wiese, Altmann & Schweinberger, 2014). Therefore, although the learning paradigm used here was designed in order to improve ecological validity in comparison with previous work, it may be that further improvements could be made regarding the selection of the identities to be learned.

In the current work, each participant was only tested once, with the delay interval varying across the sample. A necessary limitation of this design was its inability to observe within-participant memory decay and how this process may vary across individuals. An alternative method of exploring the process of forgetting, therefore, would be to utilise a within-subjects design, whereby each participant was tested at various intervals throughout the week. Although certainly a more powerful approach statistically, the issue with this procedure is that participants would be exposed to the faces during each test session. Such exposures would likely remind participants of the faces to be remembered (even if different images were used) and would therefore reinforce their memories artificially and improve accuracy on subsequent tests. Related, it is widely known that testing itself aids learning (Larsen, Butler & Roediger III, 2009), even when no feedback is given (Roediger III & Karpicke, 2006; for a review, see Roediger III & Butler, 2011). As a result, simply testing participants throughout the week would artificially increase their performance and prevent the typical process of forgetting. In order to track forgetting over multiple timepoints for a single individual, a different paradigm may be required.

In sum, this experiment adds to the sparse literature on the longitudinal process of forgetting faces. Across seven days, I found that the majority of forgetting took place in the first 24 hours, with no significant detriment after that period. In addition, self-reported face recognition ability, and to a lesser extent personality, was predictive of task performance, and these associations remained unchanged across delay intervals. Given that real-world forgetting takes place over much longer time periods than typical studies consider, there is a growing need for research investigating how face recognition deteriorates over the long term.

Supplemental Information

Supplemental Information 1 Summary data about both the participants and stimuli featured in the experiment

Click here for additional data file.

Supplemental Information 2 Raw data for the experiment for each condition

Click here for additional data file.

The author thanks Abi Davis for her critical comments throughout the project.

Additional Information and Declarations

Competing Interests

Author Contributions

Human Ethics

Data Availability

The authors declare there are no competing interests.

Robin S.S. Kramer conceived and designed the experiments, performed the experiments, analyzed the data, prepared figures and/or tables, authored or reviewed drafts of the paper, and approved the final draft.

The following information was supplied relating to ethical approvals (i.e., approving body and any reference numbers):

Ethical approval was granted by the University of Lincoln’s ethics committee (ID 3508).

The following information was supplied regarding data availability:

Both the summary and raw data for the experiment are available in the Supplementary Files.

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
