# Peer review of "Forgetting faces over a week: investigating self-reported face recognition ability and personality"

_PeerJ, doi:10.7717/peerj.11828_

## Round 0.1 · original submission · Major Revisions

I have now received two reviews of your manuscript. I thank the reviewers for their work. As you will read, both reviews are thoughtful and very clear, so please refer to them for details. Let me address a couple of issues that I consider crucial for a successful revision. First, I agree with both reviewers that you should work at the theoretical foundation of your paper, strengthening it and rendering it more cohesive. Second, I think you should better clarify the role of Experiment 1, either dropping it or presenting it mainly as a pre-test of the materials. Third, please provide more details on the procedure, as indicated by both reviewers.
I am looking forward to your revised manuscript.

Reviewer 1 ·

Basic reporting

Before this study is published in PeerJ, I recommend revising certain sections in text in order to improve the current structure and to clarify certain points.

Introduction

1. While you provide a comprehensive literature review on your constructs of interest (long-term face recognition, belief in ability and personality) you do not make it clear why this gap in the literature is important to address

2. Line 113 – you do not specify here that you are referring to the belief in ability (as measured by the Prosopagnosia index) being associated with performance

3. Line 163 – unclear what you mean by "ambient and unconstrained" images

Experiment 1

1. Your intention to pick out the optimal stimuli from Exp 1 to be used in Exp 2 should be known to the reader sooner (e.g. Current Study section and introduction of Exp 1).

2. Line 277 - it is an odd place to discuss your stimulus selection for Exp 2. It may be better off in the materials section of Exp 2.

3. Please make a clear distinction between the old/new test and the questionnaire in your materials descriptions (e.g. with subheadings)

4. It is unusual to have no mention of your planned analyses prior to the results section, for example, in form of a Design section, or at the very least, in the first paragraph of your results section.

Experimental design

Current Study

1. You do not explain in your introduction or anywhere else in the manuscript why it is important to examine the relationship between face recognition ability across different delays and self-reported belief in ability as well as personality. Please contextualise your study and research questions to make it clearer why this gap in the literature is important to address. This should be highlighted again in the General Discussion.

Experiment 1

1. Your intention to pick out the optimal stimuli from Exp 1 to be used in Exp 2 should be known to the reader sooner (e.g. Current study section, and introduction of Exp 1). Exp 1 is quite basic in design, where you merely make a similar observation as previous research (association between ability and belief in ability) and point out a well-established limitation (image recognition rather than identity recognition). Therefore, perhaps its role in stimuli selection for Exp 2 should be viewed as its primary purpose, and therefore should be highlighted at the start of Exp 1, not at the end.

2. Line 281 - could you provide a brief comment in text justifying your selection criteria for identities to be used in Exp 2 i.e. why “the lowest and then every ninth identity across the range of values for hit rate”?

3. If the images were extracted from videos, why not get two images for each identity to use across learning and recognition stages?

4. Why such a short exposure duration of 1500ms? Also having these briefly presented images shown one after the other, without letting participants pace themselves, seems unnecessarily demanding for a learning task. Please clarify in text this design choice, as it does not seem to reflect “typical” identity learning or familiarisation

5. How was the HK15 overall score computed and what did higher/lower score indicate? You may have mentioned it in the results, but it should be initially specified here. Please also provide the Cronbach’s alpha for the scale

6. Line 253 - How many participants were excluded for reason a) and how many for reason b)?

Experiment 2

1. Please provide all the relevant information on the TIPI scale in the materials section, i.e., item examples, response options, how scores were computed, Cronbach’s alpha scores, etc.

2. Having 10 items to measure 5 personality types feels a bit limited. Perhaps you can add a line justifying the use of such a short measure, either in the materials section or the discussion.

3. The HK15 should be included in the materials of Exp 2.

4. Line 417 - please clarify in text how you categorised old and new responses from the 1-5 scale, and clarify what you did with response 3 “I don’t know”?

Validity of the findings

Results partially replicate but also extend previous findings in this area. The studies are generally well executed with good participant samples and sound data analyses.

1. Other than having recruited participants with lower extraversion scores compared to the general population, what other explanations can you propose for the unexpected negative association between performance and extraversion?

2. Line 510 - the possible role of sleep and wakefulness in memory consolidation is an interesting observation in relation to your findings. I would, however, restructure/rephrase these last 9 lines of the paragraph to make it clearer to the reader that your findings are in line with previous research looking into sleep and wakefulness effects on memory.

3. Line 529 – Perhaps you could first specify why using a between subject design might be a limitation (other than a less powerful approach).

Additional comments

No comment

Reviewer 2 ·

Basic reporting

No comment.

Experimental design

See general comments.

Validity of the findings

See general comments.

Additional comments

This paper examined the relationship between self-reported facial recognition ability and facial recognition performance (Exp 1), as well as the rate at which individuals forget faces over one week, and whether accuracy correlates with self-reported facial recognition ability or self-reported personality traits (Exp 2). In the first experiment participants learned unfamiliar faces, and then completed an old/new recognition task immediately after using the exact same images. In the second experiment, participants filled out self-reported questionnaires of recognition ability and personality, then saw video footage of unfamiliar participants and were asked to complete an old/new recognition task using new images hours or days later. Across the two experiments, the authors found that self-reported facial recognition ability had a moderate correlation with actual facial recognition ability, and that extraversion had a small negative association with facial recognition performance. In Exp 2, recognition accuracy dropped substantially within 1 day but did not show evidence of further decline after 7 days.

Overall, the paper investigates research questions that are timely and of interest to the field. However, there are some issues that would need to be addressed before recommending publication.

MAJOR COMMENTS
The introduction sets out two aims: forgetting faces over time and the relationship between self-reported measures of ability and personality with face recognition accuracy. It’s not clear why these two things were investigated in the same paper – are they conceptually related? The only time the author links the two is lines 454-456, and only to note they don’t interact. Would you expect them to? If not, why investigate both questions at the same time? The conceptual framing of the paper needs to be more cohesive.

The author should strongly consider removing Experiment 1 from the paper. Experiment 1 uses the exact same images at learning and test. As the author notes (lines 297-303), this is widely considered bad practice because the test taps into low-level image recognition, not face recognition. The fact that the experiment “worked” (i.e. the sig correlation between perceived and actual performance) does not get around the issue. These days, there is really no justification to use such a design to investigate face recognition, and so meaningful conclusions cannot be drawn from the Exp 1 data.

There is a potential ethical issue in this study. How long did participants take to complete the experiments? The payment of US$0.75 for each part seems very low. I believe the standard/ethical rate is US$1-1.35/10mins (i.e. $6-$8/hr).

MINOR COMMENTS
Introduction
• The introduction focuses mainly on forgetting over time, so it comes as a surprise that Exp 1 only focuses on the relationship between perceived and actual face recognition ability.
• Lines 36-42: The first paragraph of the introduction is very clunky. The concept of face familiarity as a continuum and the difference between familiar and unfamiliar face processing are two separate ideas and the link between them is unclear.

Results: Experiment 1
• Lines 277-288: It’s unclear why this section is in the Exp 1 results, before the discussion of Exp 1 findings, and before the reader knows what Exp 2 is. The author should move this to the Method section of Exp 2.

Method: Experiment 2
• Were participants asked about the languages they speak? Although the author tested English native-speaking countries and confirmed participants’ ethnicity, this does not necessarily mean they speak English as their first and/or only language.
• Was the author able to identify which participants had sound turned on during the learning task. Could this have impacted the findings?
• Why were the HK11 and personality questionnaires administered before assessing facial recognition abilities? Could providing the questionnaires prior to face tests have alerted participants to the purpose of the experiment, or affected the results in some way?
• Lines 329-332: It should be made clear to readers that the pyMTurkR package is used to send messages to participants inviting them to complete part 2 – phrasing of “maximising likelihood” is ambiguous otherwise.
• Line 373: Was the attention check provided via audio or shown as text on-screen?
• Lines 398-399: The delay conditions are only explained here, but are referred to earlier in line 377, 362 and 333 without any explanation.
• Line 397: “logistical reasons” is unclear. Why weren’t participants randomly allocated to delay condition?
• Line 401: What does “until participation was exhausted” mean?
• Lines 406-409: Why were participants from Australia and New Zealand, but not other countries, excluded from the six-hour and twelve-hour delay conditions?

Results: Experiment 2
• Lines 438-440: More detail as to what predictors were and were not significant would be helpful. Related, it would be helpful to mention in the body of the paper what results are important to take away from Tables 2 and 3.
• Lines 452-454: This finding, as well as other correlation measures, were not clearly stated in the results section of the paper.
• Lines 441-442: It would be informative for the reader to present some measure of how well the projected learning curve fit the data points plotted.

General Discussion
• Could the authors please clarify if this work is the first demonstration of the forgetting curve for facial stimuli?
• Lines 512-513: It would be helpful to include the point in brackets in the method section.
• Lines 528-540: It’s unclear why this paragraph has been included in the discussion. Why raise an alternative design only to thoroughly explain why it’s a bad idea?
• Lines 507-508: Repeated use of “indeed”.

---

## Round 0.2 · Minor Revisions

I am happy to accept the paper, but before the final acceptance please make the tiny edits indicated by Reviewer 2.

Reviewer 1 ·

Basic reporting

no comment

Experimental design

no comment

Validity of the findings

no comment

Additional comments

no comment

Reviewer 2 ·

Basic reporting

See general comments.

Experimental design

See general comments.

Validity of the findings

See general comments.

Additional comments

The author has carefully revised the manuscript and addressed all comments. I am therefore happy to recommend publication.

The author may wish to consider the following suggestions to improve clarity:
• Line 443-444: Consider removing brackets.
• Lines 519-522: Explanation regarding “representational” and “decisional” components is unclear.
• Lines 526-528: Sentence explaining how AUC quantifies a classifier could be clearer.
• Lines 559-560: Sentence in brackets is confusing and could be removed.
• Lines 659-661: Consider removing brackets.
• Lines 681-682: Sentence regarding random distribution is a confusing justification for the design choice.
• Line 632: The author noted that extraversion may be comprised of two sub-components and may therefore make the measure difficult to interpret. Why does the author believe this? For example, does the author believe specific elements of extraversion may map onto facial recognition better than others?

---

## Round 0.3 · accepted · Accept

I am happy to inform you that your paper has been accepted for publication.